



**Big data managing in a landslide Early Warning System: experience from a ground-based**
**interferometric radar application**
Emanuele Intrieri[1], Federica Bardi[1], Riccardo Fanti[1], Giovanni Gigli[1], Francesco Fidolini[2], Nicola
Casagli[1], Sandra Costanzo[3], Antonio Raffo[3], Giuseppe Di Massa[3], Giovanna Capparelli[3], Pasquale
Versace[3].
[1] Department of Earth Sciences, University of Florence, via La Pira 4, 50121, Florence, Italy
[2] Pizzi Terra srl, via di Ripoli 207H, 50126, Florence, Italy
[3] Department of Soil Defense, University of Calabria, Ponte Pietro Bucci, Cube 41b, 87036,
Arcavacata di Rende (CS), Italy
*Correspondence to*: Emanuele Intrieri (emanuele.intrieri@unifi.it)
**Keywords**: early warning system; slope instability; big data; monitoring; landslide; risk
management; ground-based interferometric radar

## 1   Abstract

A big challenge in terms or landslide risk mitigation is represented by the increasing of the
resiliency of society exposed to the risk. Among the possible strategies to reach this goal, there is
the implementation of early warning systems. This paper describes a procedure to improve early
warning activities in areas affected by high landslide risk, such as those classified as Critical
Infrastructures for their central role in society.
This research is part of the project "LEWIS (Landslides Early Warning Integrated System): An
Integrated System for Landslide Monitoring, Early Warning and Risk Mitigation along Lifelines".
LEWIS is composed of a susceptibility assessment methodology providing information for single
points and areal monitoring systems, a data transmission network and a Data Collecting And
Processing Center (DCPC), where readings from all monitoring systems and mathematical models
converge and which sets the basis for warning and intervention activities.
In this paper we will focus on the interaction between an areal monitoring tool (a ground-based
interferometric radar) and the DCPC, and how issues such as big data transfer, real-time warning,
line of sight correction and data validation in emergency conditions have been dealt with.

## 2   Introduction

Urbanization, especially in mountain areas, can be considered a major cause for high landslide risk
because of the increased exposure of elements at risk. Among the elements at risk, important
communication routes, such as highways, can be classified as Critical Infrastructures (CIs), since
their rupture can cause chain effects with catastrophic damages on society (Geertsema et al 2009;
Kadri et al. 2014). On the other hand, modern society is more and more dependent from CIs and





their continuous efficiency (Lebaka et al., 2016), and this has risen their value over the years. The
result is a higher social vulnerability in the face of loss of continuous operation (Kröger, 2008). The
main objective was to improve the social preparedness to the growing landslide risk, according with
the suggestions of several authors (Gene Corley et al., 1998; Baldridge et al., 2011; Urlainis et al.
2014; 2015). This led to the development of several approaches and frameworks for increasing the
resiliency of society exposed to the risk (Kröger, 2008; Cagno et al., 2011 and references therein).
The resiliency policy of course involves prevention activities but also, and more importantly, those
activities needed to maintain functionality after disruption (Snyder and Burns, 2009) and to
promptly alert incoming catastrophes in order to protect people and prepare for a possible damaging
of the endangered CI. Among these activities, the implementation of integrated landslides early
warning systems (*i.e.* LEWIS, Versace et al., 2012; Costanzo et al., 2016) reveals its increasing
importance.
In this context, the methodology described in this paper has been conceived; it has been tested and
validated on a portion of an Italian highway, affected by landslides and selected as case study: it is
located in Southern Italy, along a section of the A16 highway, an important communication route
that connects Naples to Bari.
A ground based interferometer (GB-InSAR) has been installed on the test site, in order to obtain
aerial monitoring data. The installation was in an area where the only internet connection available
was 3G, with a limit of 2 gigabyte data transfer per month. Nevertheless, these data could be
managed thanks to the implemented data transmission network and Data Collecting and Processing
Center (DCPC), organized taking into account both the internet network problems and the big
amount of data produced by the interferometer.
Interferometric data are indeed complex numbers, organized in a matrix where each pixel contains
both phase and amplitude information of the backscattered signal (Bamler and Hartl, 1998;
Antonello et al., 2004); the radar employed produced a 1001x1001 complex matrix (corresponding
to ~7 megabytes) every 5 minutes. Therefore, there was the need to reduce the massive data flow
produced by the radar. For this reason data were locally and automatically elaborated in order to
produce, from a complex matrix, a simple ASCII grid containing only the pixel by pixel
displacement value, which is derived from the phase information. Then, since interferometry only
measures the displacement component projected along the radar line of sight, data needed to be re-
projected. This was performed by dividing the ASCII grid by a correction matrix, where every
element of the matrix was the percentage of the actual displacement that was measurable by the
radar; such percentage can be obtained with trigonometrical arguments knowing the position of the
radar and the direction of movement of the landslides (which, in our case, corresponded with the
slope direction) thus enabling the calculation of the radar line of sight.
To further reduce the size of the grids, matrixes where cropped in order to contain only those pixels
where relevant information could be extracted.
The ASCII grids where also averaged to reduce noise, so 8-hours and 24-hours averaged grids were
obtained. According to the early warning procedures that were defined, during periods characterized
by low or null slope movement, only 8-hours and 24-hours data where transferred, together with the
last displacement measurement of a reduced number of control points.
The transfer was performed after transforming the grids into strings and by sending them through a
middleware to the Data Acquisition and Elaboration Centre, where control points displacement





values where compared with warning thresholds and the grids where projected on a GIS
environment as 2D displacement maps.
## 3   Materials and methods
*3.1 GB-InSAR*
The ground-based interferometric synthetic aperture radar (GB-InSAR) is composed of a
microwave transceiver mounted on a linear rail (Tarchi et al., 1997; Rudolf et al., 1999; Tarchi et
al., 1999). The system used is based on a Continuous Wave – Stepped Frequency radar, which
moves along the rail at millimeter steps, in order to perform the synthetic aperture; the longer the
rail the higher the cross-range resolution. The microwave transmitter produces, step-by-step,
continuous waves around a central frequency, which influences the cross-range resolution and
determines the interferometric sensitivity i.e. the minimum measurable displacement, usually
largely smaller than the corresponding wavelength.
The radar produces complex radar images containing the information relative to both phase and
amplitude of the microwave signal backscattered by the target. The amplitude of a single image
provides the radar reflectivity of the scenario at a given time, while the phase of a single image is
not usable. The technique that enables to retrieve displacement information is called interferometry
and requires the phase from two images. In this way it is possible to elaborate a displacement map
relative to the elapsed time between the two acquisitions.
The main added value of GB-InSAR is its capability of blending the boundary between mapping
and monitoring, by computing 2D displacement maps in near real-time. The use of this tool to
monitor structures, landslides, volcanoes, sinkholes is largely documented (Calvari et al., 2016; Di
Traglia 2014; Intrieri et al., 2015; Bardi et al., 2016, 2017; Martino and Mazzanti, 2014; Severin,
2014; Tapete et al., 2013), as well as for early warning and forecasting (Intrieri et al., 2012; Carlà et
al., 2016a; 2016b; Lombardi et al., 2016).
GB-InSAR systems probably reveal their full potential in emergency conditions. They are
transportable and only require from few tens of minutes to few hours to be installed (depending on
the logistics of the site). Moreover, they are able to detect "near-real time" area displacements,
without accessing the unstable area, 24h and in all weather conditions (Del Ventisette et al., 2011;
Luzi, 2010; Monserrat et al., 2014). On the other hand, some limitations reduce the GB-InSAR
technique applicability: first of all the scenario must present specific characteristics in order to
reflect microwave radiations, maintaining high coherence values (Luzi, 2010; Monserrat et al.,
2014); only a component of the real displacement vector can be identified (i.e. the component
parallel to the sensor's line of sight); maximum detectable velocities are connected to the time that
the system needs to obtain two subsequent acquisitions. Sensors need power supply that, for long
term monitoring, cannot be replaced by batteries, generators or solar panels.
With the specific aim of performing an early warning system, data acquired *in situ* must be sent
automatically to a "control center" where they are integrated in a complete early warning system
procedure (Intrieri et al., 2013). In this sense, another main limitation is represented by the necessity
to transfer a high quantity of data, whose weight has to be reduced to the minimum, in order to
reduce the load on transmission network.



The employed system is a portable device designed and implemented by the Joint Research Center
(JRC) of the European Commission and its spin-off company Ellegi-LiSALab (Tarchi et al., 2003;
Antonello et al., 2004). The linear rail is 210 cm long, allowing a synthetic aperture of maximum
180 cm. It is easily transported by a normal motorized vehicle but its length is enough to obtain
high cross-resolution images, also at a distance from the scenario the can reach more than 1 km.
The transmitter and receiver move on the rail on a specific support, which can be tilted in order to
direct the microwave radiation as much as possible parallel to the displacement direction. The
power base represents the control center of the system; it contains a personal computer that
manages the definition of input parameters used in the acquisition phase and a UPS (Uninterruptible
Power Supply) to guarantee constant electric supply; it also enables data elaboration and storage,
being equipped with two boards increasing the memory to 1.8 TiB. The employed system needs 850
W, 230 VAC and 50 Hz as power supply. The integrated UPS guarantees the continuity of the
electric supply in case of necessity, for a period of maximum 12 hours after the power cut. The total
weight of the instrument is 95 kg, equally distributed over the different components (power base
with UPS and boards, transceiver, linear rail).
*3.2 Early warning system architecture*
Morphological features, hydrogeological factors and sudden rainfall can cause different types of
movements or fall of earthy and rock materials. The unpredictability and diversity of these events
make structural interventions often inappropriate to reduce the related risk, and real time monitoring
network difficult to implement.
In the last decade, Wireless Sensor Networks (WSNs) have been largely used in various fields. A
significant increase in the use of WSN, due to their simplicity, low cost of installation,
manufacturing and maintenance, has been recorded in the framework of environmental monitoring
applications (Intrieri et al., 2012; Liu et al., 2007; Yoo et al., 2007). Different types of sensor nodes
of these networks, distributed with high density in the monitored areas, send environmental
information to the concentrators nodes, generating a considerable amount and a wide variety of
collected data. Due to the significant growth of data volumes to be transferred, the WSN require
flexible ad-hoc protocols, able to respect constraints related to energy consumption management
(Hadadian and Kavian, 2016; Khaday et al., 2015; Parthasarathy et al., 2015). In particular, many
protocols have been developed that offer data aggregation patterns to optimize the sensor nodes
battery life (Kim et al., 2015) or sleep/measurement/data transfer cycles to minimize the energy
consumption (Fei et al., 2013; Venkateswaran and Kennedy, 2013).
LEWIS (Costanzo et al., 2016) uses heterogeneous sensors, distributed in the risk areas, to monitor
the several physical quantities related to landslides. The measured data, through a
telecommunications network, flow into the Data Collecting And Processing Center (DCPC), where,
using suitable mathematical models for the monitored site, the risk is evaluated and eventually the
state of alert for mitigation action is released (Figure 1).
The system, through a modular architecture exploiting a telecommunication network (called
LEWARnet) based on an ad-hoc communication protocol and an adaptive middleware, has a high
flexibility, which allows for the use of different interchangeable technological solutions to monitor
the parameters of interest.




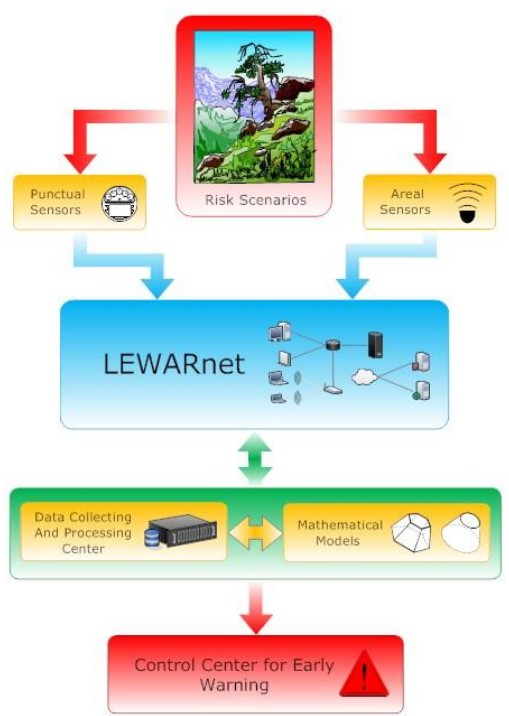


**Figure 1. LEWIS architecture.**
The telecommunication network has been designed and implemented within the Microwave
Laboratory of the University of Calabria.
The areas to be monitored have been divided into geomorphological units, indicated as GU. Each
GU is then subdivided into more heterogeneous sensors subnets.
For this purpose, each GU contains a concentrator node, called first level Sink, which has the
purpose of coordinating the sensors or any sub-network of sensors and collect the measurements
from them to transmit the data to the DCPC. Each sub-network is coordinated by a second level
Sink node, connected to the first level Sink through a short-range wireless connection (eg. ZigBee,
Bluetooth).
The network architecture can be schematized as shown in Figure 2.



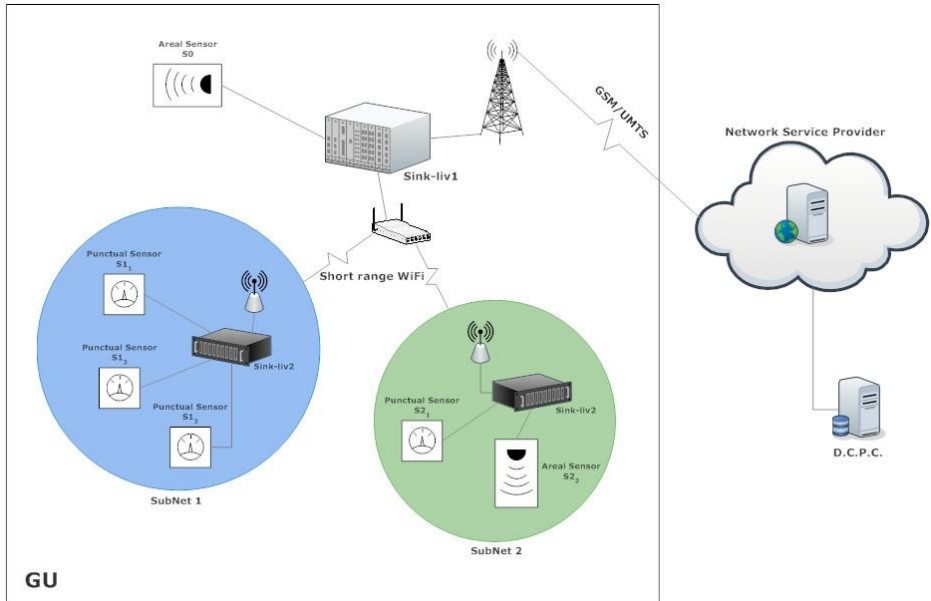


**Figure 2. LEWARnet architecture.**
The network has been equipped with both single point sensors as well as area sensors, and the sub-
networks may include the use of a single type of these or both. The present paper addresses a sub-
network comprising an area sensor, the GB-InSAR.
The different sensors types generate asynchronous traffic, thus imposing the adoption of an ad-hoc
transmission protocol. This can support an asynchronous transmission mode to the DCPC, and it is
equipped with message queues management capacity to reconstruct historical data series, between
two connection sessions, in case of null or partial transmission. This operation mode requires the
presence of a software architecture that operates as a buffer, acting as an intermediary or as
middleware (LEWARnet), between the data consumer (DCPC) and the data producers (sensors and
sub-networks of sensors).
The developed middleware also monitors the processes of transmission and data acquisition,
recognizing the activity status of the sensors and that of the DCPC, and integrating encryption and
data compression functions.

*3.3 Data Collecting and Processing Center (DCPC)*
The management of information flows, the telematic architecture and the services for data
management are entrusted to the DCPC.
The DCPC has been designed and performed according to a complex hardware and software
system, able to ensure the reliability and continuity of the service, providing advance information of
possible dangerous situations that may occur.





In the research project, the DCPC has to ensure the continuous exchange of information among
monitoring networks, mathematical models and the Command and Control Center (CCC), that is
responsible for emergency management and decision making.
The design and implementation of procedures for the exchange of information from the monitored
sites to the CCC was built according to persistent and stable communication protocols, that are
suitable for hardware/software architecture of monitoring devices, for models and for CCC.
Data flow from the monitoring network was managed according to a communication protocol,
implemented by the DCPC, and named AqSERV. AqSERV was designed considering the
heterogeneity of devices of monitoring and transmission networks (single point and area sensors)
and the available hardware resources (microcontrollers and/or industrial computers). AqSERV was
devised to link DCPC database (named LEWISDB) to the monitoring networks, after validation for
the authenticity of the node that connects to the center. Data acquisition, before the storage in the
database, is validated both syntactically and according to the information content. The procedures
for extraction of the information content and validation have been realized differently for single
point and area sensors: the latter require a more complex validation, as they work in a 2D domain.
The complete management of the monitoring networks by DCPC has been realized through specific
remote commands, sent to individual devices via AqSERV, to reconfigure the acquisition intervals
or to activate any sensor, depending on the natural phenomena occurring in real time.
The acquired and validated data are then accessible for the mathematical models through a further
service, created ad hoc, which publishes all the acquisitions by sensors on a remote server for
sharing.
The configuration of monitoring networks, composed by devices and sensors, of communication
protocol used by each network, and of rules for extraction and validation of information content is
carried out through a web application that allows for the management of the whole system by the
users.
Besides the configuration, the application has been configured to automatically create the tables of
interest; automation of the process permits to reduce the acquisition time and possible human errors.
The real-time search for acquisitions is carried out through a WebGIS, specifically designed for
WSNs, but that can be easily extended to classic monitoring networks.
The WebGIS was designed according to the traditional web architecture, client-server, by using
network services which are web mapping oriented:
- web server for static data;
- web server for dynamic data;
- server for maps;
- database for the management of map data.
The static layers provided by the WebGIS are the results produced by geological studies for the
identification of event scenarios: geological map, geomorphological map, map of event scenarios.
The dynamic layers are the acquisitions in real time by the sensors.
A DCPC operator can consult the information provided by each layer via a standard web browser
verifying the performance of the event precursors and any anomalies in acquisitions.
**4    Test site**
The test site chosen to experiment the integrated system is located in Southern Italy, along a section
of the A16 highway, an important communication route that connects Naples to Bari (Figure 3).



The A16 selected section develops in SW-NE direction, along the Southern Italian Apennine, in
correspondence with the valley of the Calaggio Creek, between the towns of Lacedonia (Campania
Region) and Candela (Puglia Region).

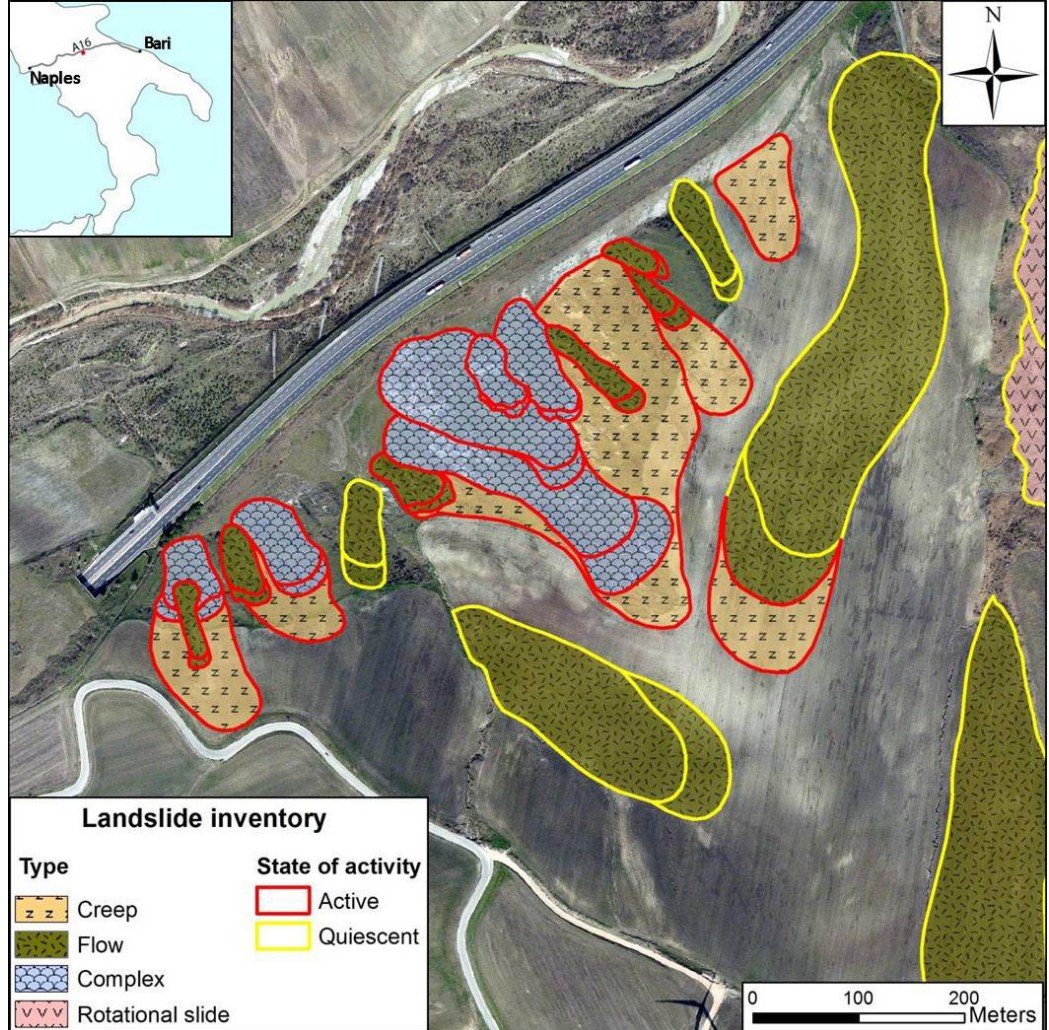


**Figure 3. Landslides detected through field survey along the monitored section of A16 highway.**
The area is tectonically active, but the landscape, characterized by gentle slopes, is mostly
influenced by lithologic factors (the strong presence of clayey sediments) rather than by tectonics.
The highway runs on the right flank of the Calaggio Creek at an altitude between 300 and 400 m
a.s.l.; the section of interest represents an element at risk in the computation of landslide risk
assessment, due to the presence of unstable areas which can potentially affect the communication
route (Figure 3).These unstable areas mainly involve clayey superficial layers.





On 1st July 2014, the GB-InSAR system has been installed on the test site. The location of the
installation point has been selected taking into account the view of the unstable area and the
distance from the power supply network. A covered structure was built in order to protect the
system from atmospheric agents and possible acts of vandalism, in the perspective of a long term
monitoring (Figure 4).

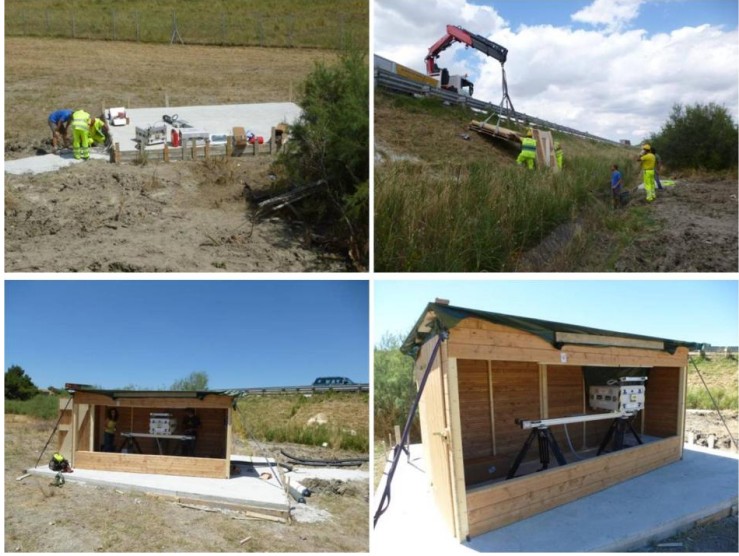


**Figure 4. Pictures of the construction of the covered structure and installation of the GB-InSAR.**
The transmission network was provided by a GSM modem, exploiting the 3G network. In addition
to the PC integrated in the GB-InSAR power base, a further external PC was exclusively employed
for data post elaboration and transmission.
The system acquired from the beginning of July 2014 until the end of July 2015.
The installation location allowed the system to detect an area between 40 and 400 meters far from
the its position in range direction, and about 360 m wide in the azimuth direction. These values,
coupled with a 40° vertical aperture of the antennas, allowed operators to detect an area of about
360 m x 360 m (Figure 5).

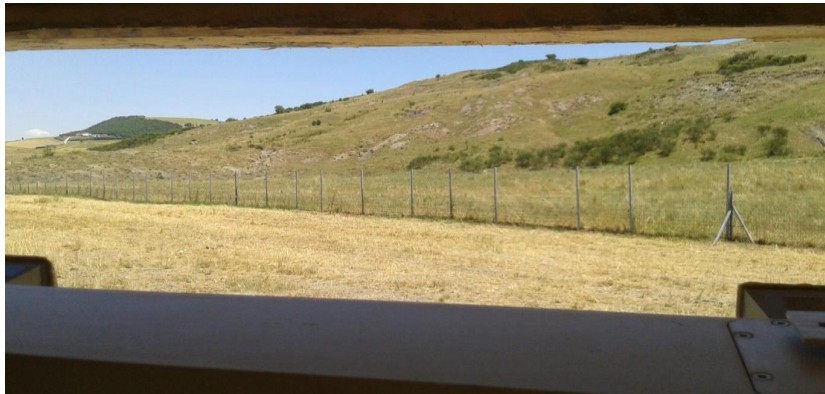






**Figure 5. Field of view of the slope from the GB-InSAR installation point.**

## 5 Data management

The most relevant matter of this monitoring was not as much related to the detection of landslide movements threatening the highway, as to how a long term monitoring performed with an instrument providing huge amounts of data could have been run without resorting to large hard drives nor to fast internet connections. For this reason an appropriate data management (Figure 6) was developed.

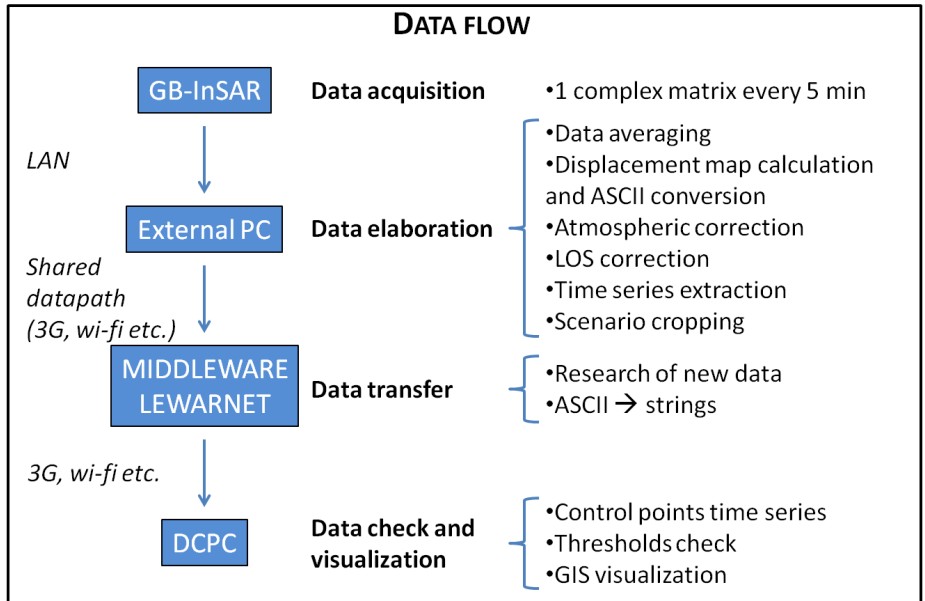

**Figure 6. Diagram showing the complete data flow from acquisition to final visualization.**

### 5.1 Data acquisition

The GB-InSAR employed produced a single radar image, consisting in a 1001x1001 complex matrix, every 5 minutes. Each one is around 8 Megabytes large, resulting in more than 2 Gigabytes of data produced every day.

This amount of data represented an issue for both store capacity and data transmission.

### 5.2 Data elaboration

After being acquired, data were then transferred through LAN connection to the external PC implementing a dedicated Matlab script locally performing the actions described as follows.





### 5.2.1 Data averaging

In order to reduce the noise normally affecting radar data (especially in vegetated areas), the images acquired every 5 minutes were also averaged using all data of the previous 8 and 24 hours. Then images averaged on 24 hours have been used to calculate daily displacement maps, every 8 hours to create 8h displacement maps and non-averaged images to calculate 5 minutes displacement maps. These time frames have been selected based on the characteristics of the slope movements and signal/noise ratio in the investigated area.

Averaging is also a mean to make a good use of a high data frequency, since it enables to reduce the memory occupied in the database as an alternative to their direct elimination.

### 5.2.2 Displacement map calculation and ASCII conversion

Each radar image can be represented as in Eq.1:

$$S_n = A_n \, exp(j\varphi_n) \tag{1}$$

where $A_n$ is the amplitude of the $n^{th}$ image, $\varphi_n$ its phase and $j = (-1)^{1/2}$ is the imaginary unit. The displacement $\Delta r$ occurred in the time period between the acquisition of $S_1$ and $S_2$ has been calculated with the following (Eq.2):

$$\Delta r = (\lambda/4\pi) \cdot \Delta\varphi \tag{2}$$

where $\lambda$ is the wavelength of the signal and

$$\Delta\varphi = \varphi_1 - \varphi_2 \tag{3}$$

can be derived from:

$$S_3 = S_1 \, S_2^* = A_1 A_2 \, exp[j(\varphi_1 - \varphi_2)] \tag{4}$$

As a result, an ASCII file, only containing the information relative to the displacement for each pixel, was obtained.

### 5.2.3 Atmospheric correction

One of the major advantages of GB-InSAR is the capability to achieve sub-millimeter precision. However this can be severely hampered by the variations of air temperature and humidity, especially when long distances are involved. Usually, atmospheric correction is performed by choosing one area considered stable, taking into account that every displacement value different from 0 is due to atmospheric noise and assuming that this offset is a linear function of the distance. Based on this relation all the displacement map is corrected. In our case the whole scenario has been selected and then only the potential unstable zones and those with a weak or incoherent backscattered signal were removed. The remaining areas were then considered stable and therefore were used for calculating the atmospheric effects. This results in a larger correction region that enables a statistical correlation between the atmospheric effects and the distance and therefore the calculation of a site-specific regression function that may not necessarily be linear (Figure 7).


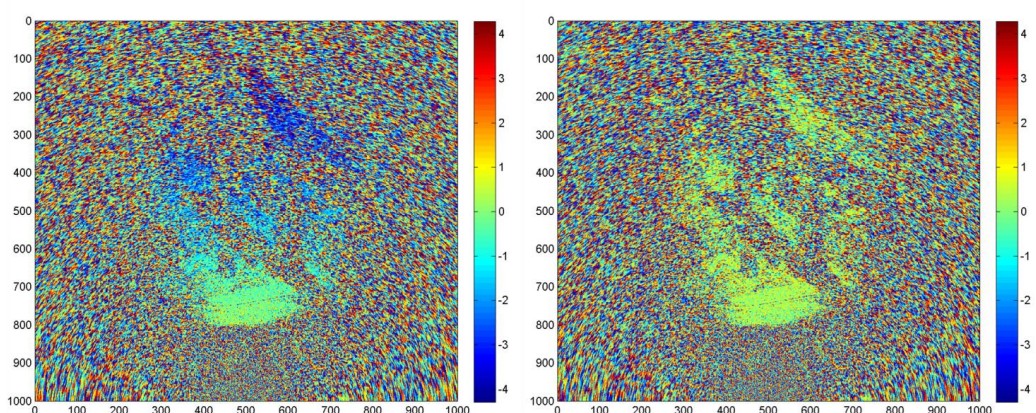

318

**Figure 7. The color bar is expressed in mm; green indicates stable pixels, while blue and red respectively movement toward and away from the GB-InSAR. Left: raw interferogram showing artificial displacement increasing linearly with distance (as typical of atmospheric noise). Right: the same interferogram after the atmospheric correction.**

### 5.2.4 Line of sight correction

The availability to detect only the LOS (Line Of Sight) component of the displacement vector represents one of the main limitations of the GB-InSAR technique. A method to partially overcome this limitation has been applied in this paper, following the procedure described in Colesanti & Wasowski, 2006 and later in Bardi et al. 2014 and 2016.

Assuming the downslope direction as the most probable displacement path, radar data have been projected on this direction. Input data as the angular values of *Aspect* and *Slope* have been derived from the *Digital Terrain Model* (DTM) of the investigated area; furthermore, azimuth angle and incidence angle of the radar LOS have been obtained.

After calculating the direction cosines of LOS and Slope (respectively functions of azimuth and incidence angles and aspect and slope angles) in the directions of Zenith ($Z_{los}$, $Z_{slope}$), North ($N_{los}$, $N_{slope}$) and East ($E_{los}$, $E_{slope}$), the coefficient $C$ is defined as follow (Eq. 5):

$$C = Z_{los} x Z_{slope} + N_{los} x N_{slope} + E_{los} x E_{slope} \qquad (5)$$

$C$ represents the percentage of real displacement detected by the radar sensor (Figure 8A).

The real displacement ($D_{real}$) is defined as the ratio between the displacement recorded along the LOS ($D_{los}$) and the $C$ value (Figure 8 B).


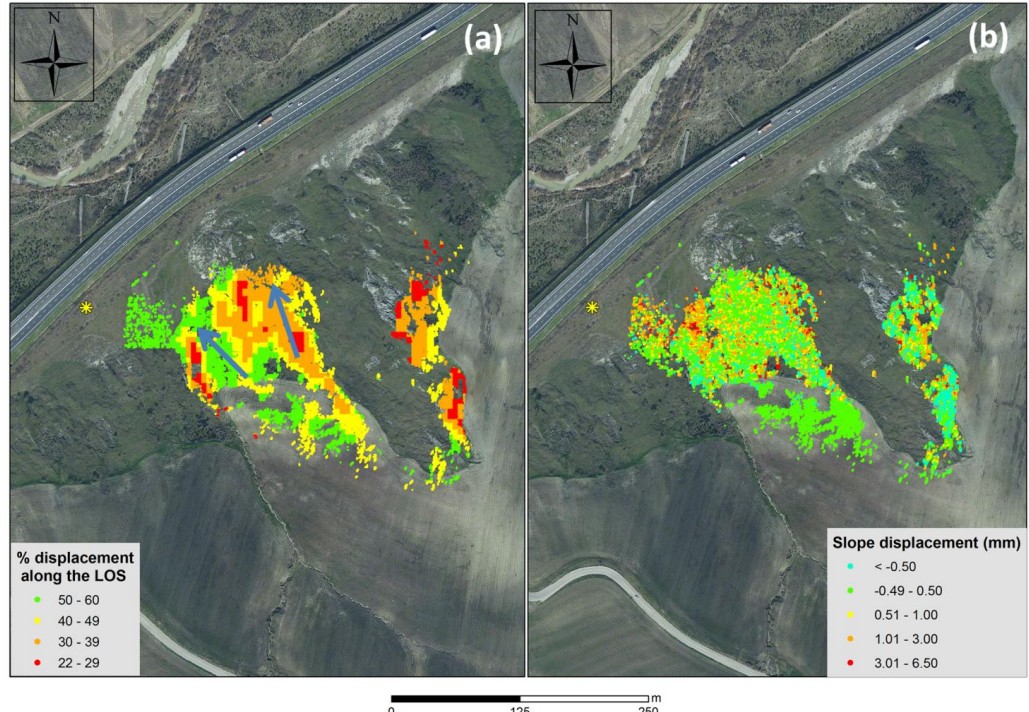


**Figure 8. (a)** *C* **values map. Blue arrows indicate the downslope direction. (b) Cumulated displacement**
**values projected along the downslope direction, referred to a period between 1 July 2014 and 1**
**November 2014.**

Assuming that the studied landslide actually moves along the downslope direction, the GB-InSAR
detectable real displacement percentage ranges between 22 and 60 % (Figure 8A).
In Figure 8B, an example of slope displacement map has been shown. Here, cumulated
displacement data related to a period between 1 July and 1 November 2014 have been projected
along the downslope direction. Data show as the area can be considered stable in the referred
period; maximum displacement values of 4 mm in 4 months (eastern portion of observed scenario)
can be still considered in the range of stability.
5.2.5   Time series extraction
In order to allow for a fast data transfer and velocity threshold comparison, some representative
control points were selected, aimed at providing cumulated displacement time series. Control points
were retrieved from the same displacement maps calculated as described in paragraph 5.2.2 and
therefore can be relative to a time frame of 5 minutes, 8 hours or 24 hours.
In case of noisy data, instead of having a time series relative to a single pixel, these can be retrieved
from a spatial average obtained from a small area consisting of few pixels.



### 5.2.6 Scenario cropping

Typically, the field of view of a GB-InSAR is larger than the actual area to be monitored. In fact, a portion of the radar image may be relative to the ground, sky, areas geometrically shadowed or covered by dense vegetation. These may be of no interest or even containing no information at all. For the case here studied around 50% of a radar image had a low coherence and was for all practical purposes, unusable. Therefore, a cropping of the ASCII displacement map occurred in order to frame only the relevant area.

### *5.3 Data transfer and visualization*

The interferometric data generated by GB-InSAR, after the pre-processing and proper correction previously described, are ready for transfer to the DCPC. The transmission of these data to the DCPC is mediated by the middleware, which interrogates the GB-InSAR for tracking the state, detects the newest data, and reorders and marks them to properly build data time series to be transferred to DCPC.

Subsequently, the middleware manages communications with the DCPC, according to the implemented ad-hoc protocol. This ensures the security of data providers through encrypted authentication mechanisms, it allows for recovering missing or partially transmitted data, thus avoiding information loss, and provides data acquired by the sensors to the DCPC in a standardized format, JSON, able to guarantee uniformity between the various information provided by the various sensors types. All these particular features fully justify the adoption of an ad-hoc protocol for data transfer, instead of using a standard protocol such as FTP.

The data files produced by the GB-InSAR have already been locally pre-processed and result in a matrix expressed in ASCII code; the dimensions of the matrix are known and range from 1x1 (for the displacement of single control points) to 1001x1001 (for uncropped displacement maps). Before encapsulating these data in the message to be transferred to DCPC, the middleware converts them from ASCII code to character strings, using the standard coding ISO / IEC 8859-1, so being able to obtain a data compression with a factor equal to $\approx 8$.

Eventually the DCPC is entrusted for cumulating the displacements relative to the control points, which are compared with the respective thresholds, and for visualizing the displacement maps as WebGIS layers, thus enabling data validation and the evaluation of the extension of moving surface.

## 6 Early warning procedures discussions

The GB-InSAR is part of a larger early warning system (LEWIS) which also includes other monitoring systems and simulation models. Therefore, to understand how GB-InSAR data can be used in an early warning perspective, it is necessary to make reference to LEWIS as a whole.
Any information, coming from the investigated sites and subsequently processed also by using the simulation models, is used to define an intervention model. This is based on the following elements: event scenarios, risk scenarios, levels of criticality, levels of alert.
Event scenarios describe the properties of expected phenomena in terms of dimension, velocity, involved material and occurrence probability. Occurrence probability depends on the associated time horizon, which should be equal to few hours at most, in the case of early warning systems.



Evaluation of occurrence probability is carried out by using information from monitoring systems
and/or from outputs of adopted mathematical models for nowcasting. All the properties, to be
analyzed for event scenarios, are listed below; a subdivision in classes is adopted for each one:
• landslide velocity (5 classes from slow to extremely rapid);
• landslide surface (5 classes from very small to very large);
• landslide scarp (5 classes from very small to very large);
• landslide volume (5 classes from extremely small to large);
• thickness (5 classes from very shallow to very deep);
• magnitude (3 classes: low, moderate, high), which combines the previous information;
• involved material (mud, debris, earth, rock, mixture of components);
• occurrence probability (zero, low, moderate, high, very high, equal to 1).
While some of the aforementioned parameters are determined by geological surveys, landslide
velocity is directly derived from monitoring data (such as those collected by GB-InSAR). Landslide
surface can be determined by geomorphological observation but is precisely quantified by GB-
InSAR, thanks to its capability of producing 2D displacement maps.
Risk scenarios can be firstly grouped in the following three classes:
A. mud and/or debris movements which could induce a friction reduction and facilitate slips;
B. road subsidence induced by landslides that could drag or drop vehicles;
C. falls of significant volumes and/or boulders that could crush or cover vehicles and constitute
an obstacle for others vehicles.
For each previous risk scenario, six sub-scenarios can be identified on the basis of the number of
potentially involved infrastructures, carriageways and lanes (a. hydraulic infrastructures and/or
barriers, b. only emergency lane, c. lane, d. fast lane, e. fast lane of the opposite carriageway, f. lane
of the opposite carriageway). Thus, all possible risk scenarios are 18 (Figure 9) , indicated with a
couple of letters (Capital and small).





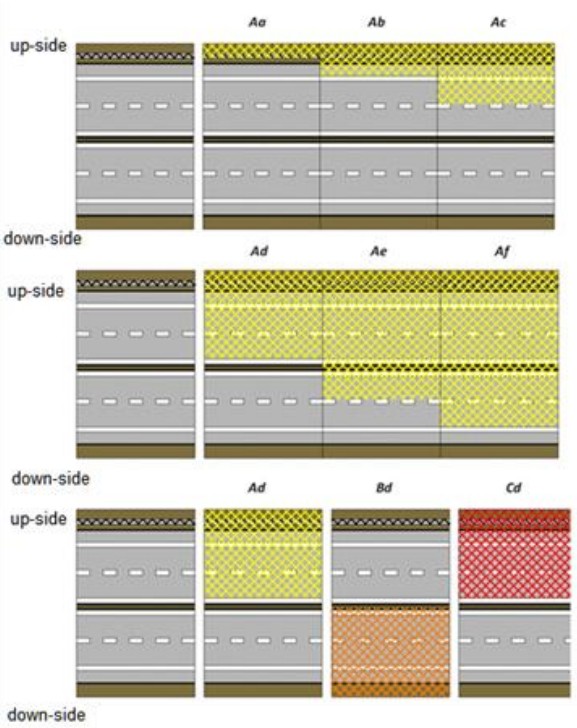

**422 Figure 9. Top and middle: possible risk scenarios involving the scenario A (landslides that could**
**423 reduce friction) to increasing sectors of the highway. Bottom: combinations of scenarios with different**
**424 types of phenomena (A, B, C) affect the emergency lane, lane and fast lane.**

The following information is provided to DCPC:
• Measurements from sensors
• Model outputs
and four states are identified for each of them:
• state 0 = no variation
• state 1 = small variation
• state 2 = moderate variation
• state 3 = high variation.
Besides information from sensors and models, other information is obtained from meteorological
and hydrological models (named as indicators).
Indicators comprise weather forecasting and output of FLaIR and Sushi models (Sirangelo et al.
2003; Capparelli and Versace 2011) on the basis of observed and predicted (for the successive six
hours) rainfall heights.
Two states are defined for indicators:
• state 0 = no variation or not significant,
• state 1 = significant variation.
To sum up, DCPC has the following information in any moment:
‣ state (0, 1) of indicators (IND),
‣ state (0, 1, 2, 3) of sensors and models running for the specific highway section (SEN),





and, on the basis of these states, four different decisions can be made by DCPC, one of which with three options.

All the possible decisions are illustrated in Table 1, in which the weight of the several sensors is assumed to be the same. Based on the notices of criticality levels provided by the DCPC, and on its own independent evaluations, the CCC issues the appropriate warning notices (Surveillance, Alert, Alarm and Warning) and makes decisions about the consequent actions.

| State of sensors and/or models | DCPC decisions |
|---|---|
| All INDs and SENs are S0 | 0 - no decision |
| At least one IND is S1 and all SENs are S0 | 1 – SOD (Sensor On Demand) activation |
| At least one SEN is S1 | 2 – to intensify the presence up to 24 hours/day |
| At least $n$ SENs are S1 or at least one SEN is S2 | 3/1 – to issue a notice of ordinary criticality (level 1) |
| At least $n$ SENs are S2 or at least one SEN is S3 | 3/2 - to issue a notice of moderate criticality (level 2) |
| At least $n$ SENs are S3 | 3/3 - to issue a notice of high or severe criticality (level 3) |

**Table 1. DCPC possible decisions.**

The information of each sensor and the results produced by the models are used to assess, in each instant, the occurrence probability of an event scenario in the monitored areas and the possible risk scenarios.

This combination of heterogeneous data was carried out by identifying for each sensor and model a typical information (displacement, precipitation, inclination, etc.), evaluating the state in each instant, according to a threshold system, and combining this result for all sensors placed in a monitored geomorphological area.

The final result is constituted by the occurrence probability of an event scenario, that is associated with a specific action by the DCPC. In particular, if the occurrence probability is low, moderate or high it is necessary to issue a notice of criticality (ordinary - Level 1, moderate - Level 2, High - Level 3) to the CCC.

The DCPC sends two types of information:

1) criticality state of the single monitored geomorphological unit,

2) criticality state of the whole area.

The adopted communication protocol between the two centers for the exchange of information was carried out through a web service provided by the CCC, using the classes and attributes of the methodology named Datex II (which is a protocol for the exchange of traffic data). The use of the web service allowed to ensure the interoperability of data between the two centers, regardless of the used hardware and software architecture, through a persistent service capable of ensuring an immediate restoration of the connections, in case of malfunction and a continuous monitoring between the two centers, even in the absence of criticality.





## 7    Conclusions

The GB-InSAR is a monitoring tool that is becoming more and more used in landslide monitoring and early warning, especially thanks to its capability of producing real-time, 2D displacement maps. On the other hand, it still suffers from some drawbacks, such as the limitation of measuring only the LOS component of a target's movement and logistic issues like those owing to a massive production of data that may cause trouble for both storing capacity and data transfer.

These problems have been addressed when a GB-InSAR was integrated within a complex early warning system (LEWIS) and only a limited internet connection was available. This situation required that a series of pre-elaboration processes and data management procedures took place in situ, in order to produce standardized and reduced files, carrying only the information needed when it was needed. The procedures mainly concerned the transmission of data averaged over determined time frames, proportionate with the kinematics of the monitored phenomenon. Before, transmission data were also corrected (both in terms of atmospheric noise and LOS) and reduced, by filtering out the information relative to the amplitude of the targets, by eliminating the areas not relevant for the monitoring and by transforming the matrices into strings.

As a result, GB-InSAR data converged into the early warning system and contributed to it by producing displacement time series of representative control points to be compared with fixed thresholds. Displacement maps were also available for data validation by expert operators and for retrieving information relative to the surface of the moving areas.

*Competing interests*. The authors declare that they have no conflict of interest.

*Acknowledgements*. This research is part of the project "LEWIS (Landslides Early Warning Integrated System): An Integrated System for Landslide Monitoring, Early Warning and Risk Mitigation along Lifelines", financed by the Italian Ministry of Education, Universities and Research and co-funded by the European Regional Development Fund, in the framework of the National Operational Programme 2007-13 "Research and Competitiveness", grant agreement no. PON01_01503.
The Authors are thankful to Giuseppe Della Porta and his colleagues from Autostrade S.p.A. for their availability in permitting and supporting the installation and maintenance of the GB-InSAR along the A16 highway.

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
