# Peer review of "Big data managing in a landslide Early Warning System: experience from a ground-based"

_Natural Hazards and Earth System Sciences, 2017_

## Referee Comment (RC1) · Anonymous Referee #1 · 3 Aug 2017

General comments The paper presents a procedure for the integration of GB-InSAR data within an early warning integrated system for risk prevention for a critical infrastructure (A16 highways connecting Naples to Bari in southern Italy). The use of GB-InSAR for landslide monitoring is not new in the scientific literature, although not yet standardized, so its use in EWS is certainly of interest to the community of landslide researchers. The used language is correct and readable. However, some changes are suggested before publication on NHESS journal. The weak point is that the current version of the paper appears as a "technical note" rather than an original research article. Indeed, the Authors provide plenty of details concerning the LEWIS system (that is not central in the work) and both the installation and set-up of GB-InSAR tool

but, on the other hand, the interpretation of the results and the adoption of thresholds for early warning purposes based on GB-InSAR data is not given the same room and relevance. So, given the timely topic addressed and the potential of these kind of applications, Authors are invited to better balance different parts of the paper to improve the overall quality and readability of the work for the typical audience of NHESS. Some suggestion are provided hereafter.

Specific comments Lines 55 to 83 provide too many details anticipating the technical descriptions that are expected instead in section 3 or 5. Please remove from here. In section 3.1 the description of LEWIS should be reduced since the Authors already refer to the published work of Costanzo et al. (2016). In section 6, please better clarify how GB-InSAR data interpretation and analysis contribute to fix thresholds for early warning. Technical corrections: - In the abstract do not use future tense (line 29) - Lines 76, 78: change "where" in "were". - Figure 1: change the shaded fonts because they are not readable - Figure 2: increase the font size. - Lines 251, 252: use the past tense. - Line 327: add references to: - Cascini et al., 2010 (for first maps of DInSAR data projected along the steepest slope direction). Cascini L., Fornaro G., Peduto D. (2010). Advanced low- and full-resolution DInSAR map generation for slow-moving landslide analysis at different scales. Engineering Geology, 112 (1-4), 29-42, doi:10.1016/j.enggeo.2010.01.003.; and to Cascini, L., Peduto, D., Pisciotta, G., Arena, L., Ferlisi, S., and Fornaro, G. (2013): The combination of DInSAR and facility damage data for the updating of slow-moving landslide inventory maps at medium scale, Nat. Hazards Earth Syst. Sci., 13, 1527-1549, doi:10.5194/nhess-13-1527-2013, for the map of projectable DInSAR data. - Line 412: please clarify better to which "friction" you are referring.

---

## Referee Comment (RC2) · Anonymous Referee #2 · 21 Aug 2017

The authors submitted a work to present the role of GB-InSAR on an integrated system for landslide monitoring. In particular, the early warning system architecture and the data management are treated. The work is referred to the LEWIS project and it is focused on a critical infrastructure in southern Italy (A16 highways).

Although the technology here presented is now well-known, the integration with other monitoring technique and the development of EWS are interesting topics.

The objectives of the manuscript are clear and the paper covers an area of interest to the journal's readership.

In order to improve the manuscript, I recommend authors to summarize the section with

irrelevant details for readers (eg. GB-InSAR features..) focusing on data analysis and interpretation, also by adding some displacement time series. In addition, it is really important to improve the conclusions, also by focusing on data integration for EWS.

Following, some specific comments and minor points to improve the text:

- Page 2, row 56: please replace "aerial" with "spatial"

- Page 2, rows 56-57: please replace "The installation was in an area where the only internet connection available 57 was 3G" with "the monitoring area was covered by a 3G mobile telecommunication networks"

- Page 2, rows 61-83: these lines are very specific and of little interest to readers. Please consider deleting these lines or inserting them in Section 3.3.

- Page 2, row 76: please change "where" with "were"

- Page 3, row 86: please change "ground-based interferometric synthetic aperture radar" with "Ground-Based Interferometric Synthetic Aperture Radar (GB-InSAR)"

- Section 3.1: this section appears too long. Please consider reducing sentences and adding a table with the technical specifications of equipment used (eg. Frequency, Bandwidth, Range and Cross-range resolution, etc.).

- Section 4: please add more geological information (eg. materials involved) to better frame the area under study.

- Page 8, row 250: please add space before "These"

- Fig. 1: please improve the quality of figure

- Fig. 3: please increase the font size

- Fig. 8: please insert the location of GB-InSAR instrument installed.
* * *
2017-178, 2017.

---

## Editor Comment (EC1) · S. L. Gariano (Editor) · 21 Aug 2017

Dear Authors,

both reviewers considered your work good and publishable, after revisions. In particular, they believe that the "technical" parts of the work should be reduced and/or summarized.

I agree with them. Thus, I have also some suggestions, listed below:

- Text in rows 61-83 could be largely reduced.

- Sections 3.1, 3.2, and 3.3 could be reduced.

[Figure]

- Figures 4 and 5 could be deleted.

- Finally, I suggest reviewing the abstract by adding more precise information about the method described in the paper and the obtained results.

Sincerely,

Stefano Luigi Gariano

---

## Author Comment (AC1) · 22 Aug 2017

Answers to Anonymous Referee #1 General comments The paper presents a procedure for the integration of GB-InSAR data within an early warning integrated system for risk prevention for a critical infrastructure (A16 highways connecting Naples to Bari in southern Italy). The use of GBInSAR for landslide monitoring is not new in the scientific literature, although not yet standardized, so its use in EWS is certainly of interest to the community of landslide researchers. The used language is correct and readable. However, some changes are suggested before publication on NHESS journal. The weak point is that the current version of the paper appears as a "technical note" rather than

an original research article. Indeed, the Authors provide plenty of details concerning the LEWIS system (that is not central in the work) and both the installation and set-up of GB-InSAR tool but, on the other hand, the interpretation of the results and the adoption of thresholds for early warning purposes based on GB-InSAR data is not given the same room and relevance. So, given the timely topic addressed and the potential of these kind of applications, Authors are invited to better balance different parts of the paper to improve the overall quality and readability of the work for the typical audience of NHESS. Some suggestion are provided hereafter. The paper will be submitted as technical note, as suggested by the reviewer. The paper has been balanced following the suggestions furnished in the following comments of the reviewer. In particular, we have better explained the method used here to set thresholds but, since the setting of thresholds is not the objective of the paper, we have also explained that the system is open and different methods can be implemented as well. Furthermore, the part concerning LEWIS has been reduced and a figure removed, in order to better balance the topic of the paper.

Specific comments Lines 55 to 83 provide too many details anticipating the technical descriptions that are expected instead in section 3 or 5. Please remove from here. This paragraph has been in part moved in paragraph 5 and in large part deleted since it mainly anticipated concepts more deeply described in paragraph 5. In section 3.1 the description of LEWIS should be reduced since the Authors already refer to the published work of Costanzo et al. (2016). The section containing information about LEWIS (3.2) has been reduced as suggested and in particular a figure has been removed (Fig. 2). Only the parts that are important to allow the reader to easily understand the following sections have been kept. In section 6, please better clarify how GB-InSAR data interpretation and analysis contribute to fix thresholds for early warning.

Technical corrections: - In the abstract do not use future tense (line 29) – Done Lines 76, 78: change "where" in "were". This part has been removed. - Figure 1: change the shaded fonts because they are not readable The font has been changed and the

shaded box now have a solid colour. - Figure 2: increase the font size. This figure has been removed. - Lines 251, 252: use the past tense. Done. - Line 327: add references to: - Cascini et al., 2010 (for first maps of DInSAR data projected along the steepest slope direction). Cascini L., Fornaro G., Peduto D. (2010). Advanced low- and full-resolution DInSAR map generation for slowmoving landslide analysis at different scales. Engineering Geology, 112 (1-4), 29-42, doi:10.1016/j.enggeo.2010.01.003.; and to Cascini, L., Peduto, D., Pisciotta, G., Arena, L., Ferlisi, S., and Fornaro, G. (2013): The combination of DInSAR and facility damage data for the updating of slow-moving landslide inventory maps at medium scale, Nat. Hazards Earth Syst. Sci., 13, 1527-1549, doi:10.5194/nhess-13-1527-2013, for the map of projectable DInSAR data. The references to Cascini et al. have been added. - Line 412: please clarify better to which "friction" you are referring. We were referring to the friction between vehicles and the tar. Now it is specified in the text.

Please also note the supplement to this comment:
https://www.nat-hazards-earth-syst-sci-discuss.net/nhess-2017-178/nhess-2017-178-AC1-supplement.pdf
* * *

---

## Author Comment (AC2) · 22 Aug 2017

Answers to Anonymous Referee #2 The authors submitted a work to present the role of GB-InSAR on an integrated system for landslide monitoring. In particular, the early warning system architecture and the data management are treated. The work is referred to the LEWIS project and it is focused on a critical infrastructure in southern Italy (A16 highways). Although the technology here presented is now well-known, the integration with other monitoring technique and the development of EWS are interesting topics. The objectives of the manuscript are clear and the paper covers an area of interest to the journal's readership. In order to improve the

manuscript, I recommend authors to summarize the section with irrelevant details for readers (eg. GB-InSAR features..) focusing on data analysis and interpretation, also by adding some displacement time series. The sections highlighted by the referee have been heavily reduced and 3 figures have been removed in total. Concerning the time series, we have not included them since these are not really meaningful. In fact, the slope did not experience significant movements and the growth of the vegetation produces noise that concealed the slightest deformations, which did not exceed the instrumental resolution (less than one millimiter per day). On the other hand, displacement maps (now figure 7) have been included, in order to show some displacement data nonetheless and to show that displacements were negligible. Indeed, as you rightly pointed put, the use of GB-InSAR for landslide monitoring is not new and it was not the aim of our paper. Our scope was to explain a procedure to overcome some logistic issues encountered in an early warning system, such as big data. We have added to better explain our scope in the abstract and in the introduction. They are reported below. In the abstract: "The aim of this paper is to show how logistic issues linked to advanced monitoring techniques such as big data transfer and storing, can be dealt with, compatibly with an early warning system. Therefore, we focus on the interaction between an areal monitoring tool (a ground-based interferometric radar) and the DCPC. By converting complex data into ASCII strings and through appropriate data cropping and average, and by implementing an algorithm for line of sight correction, we managed to reduce the data daily output without compromising the capability of performing." In the introduction: "One of the main drawbacks of advanced instruments such as GB-InSAR is how to handle the large data flow deriving from continuous real-time monitoring. The issue is to reduce the capacity needed for analyzing, transmitting and storing big data without losing important information. The main feature of this paper is indeed the management of monitoring data in order to filter, correct, transfer and access them compatibly with the needs of an early warning system." In addition, it is really important to improve the conclusions, also by focusing on data integration for EWS. Thank you for pointing out this issue. We improved the

conclusions by better explaining the possible usefulness of our paper with reference to similar situations. Unfortunately, we do not have data from other instruments; in fact, all the monitoring devices were independent and the integration was only needed at a higher level, when monitoring data and results from modeling were finally integrated and a risk assessment was possible. These aspects are already treated in other paper cited in the manuscript (Versace et al., 2012; Costanzo et al., 2015; 2016). In this paper, we only deal with the interaction between GB-InSAR and the DCPC. In fact, what happen next (e.g. data integration between GB-InSAR and other instruments) falls out of our interest and detailed knowledge. Following, some specific comments and minor points to improve the text: - Page 2, row 56: please replace "aerial" with "spatial" Done. - Page 2, rows 56-57: please replace "The installation was in an area where the only internet connection available 57 was 3G" with "the monitoring area was covered by a 3G mobile telecommunication networks" This sentence has been changed as suggested and it is now at the beginning of section 5, following the suggestion of referee #1. - Page 2, rows 61-83: these lines are very specific and of little interest to readers. Please consider deleting these lines or inserting them in Section 3.3. They have been deleted and only in small part moved. - Page 2, row 76: please change "where" with "were" This part has been removed from the paper. - Page 3, row 86: please change "ground-based interferometric synthetic aperture radar" with "Ground-Based Interferometric Synthetic Aperture Radar (GB-InSAR)" We have now changed this. - Section 3.1: this section appears too long. Please consider reducing sentences and adding a table with the technical specifications of equipment used (eg. Frequency, Bandwidth, Range and Cross-range resolution, etc.). Thank you for this comment. In fact this part was mostly a repetition and has been largely reduced. We also remove the part explaining the technical specifications of the equipment, since this information can be found in the literature cited in the paper and is not fundamental for our purpose. - Section 4: please add more geological information (eg. materials involved) to better frame the area under study. Now we have explained that "The lithologies outcropping in this area are Pliocene-Quaternary clay, clayey marlstones,

and more recent (Holocene) terraced alluvial sediments (from clay to gravel). The landslides shown in Figure 2 are all located in clay or clayey marlstones". - Page 8, row 250: please add space before "These" Done. - Fig. 1: please improve the quality of figure Now the text is bolder and the boxes are no longer filled with a gradient but with a solid color. - Fig. 3: please increase the font size The font size has been increased. - Fig. 8: please insert the location of GB-InSAR instrument installed. You are right. The yellow asterisk in the left of the images represents the location of the GB-InSAR. This is now specified in the caption of the image.

Please also note the supplement to this comment:
https://www.nat-hazards-earth-syst-sci-discuss.net/nhess-2017-178/nhess-2017-178-AC2-supplement.pdf

**Supplement:**

**Dear Editor and Referees,**

**We wish to thank you for your effort in improving our paper. We agree that several parts of the manuscript needed to be reviewed. In particular we feel that now this work is more incisive and its purpose clearer. This was thanks to your suggestions, especially those relative to cutting unnecessary parts or to better explaining some others.**

**Following, we resume all the answers to your requests together in this single file because they are tightly linked to each other and we think that in this way they are more understandable.**

**In order to give informed answer, we preferred to modify the manuscript according to your revisions, although this was not necessary at this stage (this is why we could not upload it yet). In this way, we were able to answer your suggestions not just theoretically but according to what we actually changed in the manuscript.**

**Best regards on behalf of all co-authors,**

**Emanuele Intrieri**

**Answers to S. L. Gariano (Editor)**

Dear Authors, both reviewers considered your work good and publishable, after revisions. In particular, they believe that the "technical" parts of the work should be reduced and/or summarized. I agree with them. Thus, I have also some suggestions, listed below:

- Text in rows 61-83 could be largely reduced.

**It has been almost entirely deleted (see also answers to Referee #1 and #2).**

- Sections 3.1, 3.2, and 3.3 could be reduced.

**Section 3.1: the part relative to how interferometric data are made and how they are elaborated has been deleted (see also answers to Referee #1 and #2).**

**Section 3.2: this has been reduced and a figure has been deleted (see also answers to Referee #1).**

**Section 3.3: this has been reduced as suggested.**

- Figures 4 and 5 could be deleted.

**They have been deleted from section 4. For reference, in the new version their numbering was no more 4 and 5 but 3 and 4.**

- Finally, I suggest reviewing the abstract by adding more precise information about the method described in the paper and the obtained results.

Sincerely, Stefano Luigi Gariano

**Thank you for your observation. In fact we probably missed to properly convey the message of our paper. This might have created confusion in some of the comments of the two referees concerning the fact that our aim and obtained results are not the monitoring data themselves, rather the procedures employed to obtain them. Therefore we changed the final part of the abstract ad replaced it with the following sentence:**

**"The aim of this paper is to show how logistic issues linked to advanced monitoring techniques such as big data transfer and storing, can be dealt with, compatibly with an early warning system. Therefore, we focus on the interaction between an areal monitoring tool (a ground-based interferometric radar) and the DCPC. By converting complex data into ASCII strings and through appropriate data cropping and average, and by implementing an algorithm for line of sight correction, we managed to reduce the data daily output without compromising the capability of performing".**

**Answers to Anonymous Referee #1**

General comments

The paper presents a procedure for the integration of GB-InSAR data within an early warning integrated system for risk prevention for a critical infrastructure (A16 highways connecting Naples to Bari in southern Italy). The use of GBInSAR for landslide monitoring is not new in the scientific literature, although not yet standardized, so its use in EWS is certainly of interest to the community of landslide researchers. The used language is correct and readable. However, some changes are suggested before publication on NHESS journal. The weak point is that the current version of the paper appears as a "technical note" rather than an original research article. Indeed, the Authors provide plenty of details concerning the LEWIS system (that is not central in the work) and both the installation and set-up of GB-InSAR tool but, on the other hand, the interpretation of the results and the adoption of thresholds for early warning purposes based on GB-InSAR data is not given the same room and relevance. So, given the timely topic addressed and the potential of these kind of applications, Authors are invited to better balance different parts of the paper to improve the overall quality and readability of the work for the typical audience of NHESS. Some suggestion are provided hereafter.

**The paper will be submitted as technical note, as suggested by the reviewer. The paper has been balanced following the suggestions furnished in the following comments of the reviewer. In particular, we have better explained the method used here to set thresholds but, since the setting of thresholds is not the objective of the paper, we have also explained that the system is open and different methods can be implemented as well. Furthermore, the part concerning LEWIS has been reduced and a figure removed, in order to better balance the topic of the paper.**

 Specific comments

Lines 55 to 83 provide too many details anticipating the technical descriptions that are expected instead in section 3 or 5. Please remove from here.

**This paragraph has been in part moved in paragraph 5 and in large part deleted since it mainly anticipated concepts more deeply described in paragraph 5.**

 In section 3.1 the description of LEWIS should be reduced since the Authors already refer to the published work of Costanzo et al. (2016).

**The section containing information about LEWIS (3.2) has been reduced as suggested and in particular a figure has been removed (Fig. 2). Only the parts that are important to allow the reader to easily understand the following sections have been kept.**

In section 6, please better clarify how GB-InSAR data interpretation and analysis contribute to fix thresholds for early warning.

Technical corrections: - In the abstract do not use future tense (line 29) –

**Done**

Lines 76, 78: change "where" in "were".

**This part has been removed.**

- Figure 1: change the shaded fonts because they are not readable

**The font has been changed and the shaded box now have a solid colour.**

 - Figure 2: increase the font size.

**This figure has been removed.**

- Lines 251, 252: use the past tense.

**Done.**

- Line 327: add references to:

- Cascini et al., 2010 (for first maps of DInSAR data projected along the steepest slope direction). Cascini L., Fornaro G., Peduto D. (2010). Advanced low- and full-resolution DInSAR map generation for slowmoving landslide analysis at different scales. Engineering Geology, 112 (1-4), 29-42, doi:10.1016/j.enggeo.2010.01.003.;

and to Cascini, L., Peduto, D., Pisciotta, G., Arena, L., Ferlisi, S., and Fornaro, G. (2013): The combination of DInSAR and facility damage data for the updating of slow-moving landslide inventory maps at medium scale, Nat. Hazards Earth Syst. Sci., 13, 1527-1549, doi:10.5194/nhess-13-1527-2013, for the map of projectable DInSAR data.

**The references to Cascini et al. have been added.**

- Line 412: please clarify better to which "friction" you are referring.

**We were referring to the friction between vehicles and the tar. Now it is specified in the text.**

**Answers to Anonymous Referee #2**

The authors submitted a work to present the role of GB-InSAR on an integrated system for landslide monitoring. In particular, the early warning system architecture and the data management are treated. The work is referred to the LEWIS project and it is focused on a critical infrastructure in southern Italy (A16 highways). Although the technology here presented is now well-known, the integration with other monitoring technique and the development of EWS are interesting topics. The objectives of the manuscript are clear and the paper covers an area of interest to the journal's readership.

In order to improve the manuscript, I recommend authors to summarize the section with irrelevant details for readers (eg. GB-InSAR features..) focusing on data analysis and interpretation, also by adding some displacement time series.

**The sections highlighted by the referee have been heavily reduced and 3 figures have been removed in total. Concerning the time series, we have not included them since these are not really meaningful. In fact, the slope did not experience significant movements and the growth of the vegetation produces noise that concealed the slightest deformations, which did not exceed the instrumental resolution (less than one millimiter per day). On the other hand, displacement maps (now figure 7) have been included, in order to show some displacement data nonetheless and to show that displacements were negligible. Indeed, as you rightly pointed put, the use of GB-InSAR for landslide monitoring is not new and it was not the aim of our paper. Our scope was to explain a procedure to overcome some logistic issues encountered in an early warning system, such as big data. We have added to better explain our scope in the abstract and in the introduction. They are reported below.**

**In the abstract: "The aim of this paper is to show how logistic issues linked to advanced monitoring techniques such as big data transfer and storing, can be dealt with, compatibly with an early warning system. Therefore, we focus on the interaction between an areal monitoring tool (a ground-based interferometric radar) and the DCPC. By converting complex data into ASCII strings and through appropriate data cropping and average, and by implementing an algorithm for line of sight correction, we managed to reduce the data daily output without compromising the capability of performing."**

**In the introduction: "One of the main drawbacks of advanced instruments such as GB-InSAR is how to handle the large data flow deriving from continuous real-time monitoring. The issue is to reduce the capacity needed for analyzing, transmitting and storing big data without losing important information. The main feature of this paper is indeed the management of monitoring data in order to filter, correct, transfer and access them compatibly with the needs of an early warning system."**

In addition, it is really important to improve the conclusions, also by focusing on data integration for EWS.

**Thank you for pointing out this issue. We improved the conclusions by better explaining the possible usefulness of our paper with reference to similar situations. Unfortunately, we do not have data from other instruments; in fact, all the monitoring devices were independent and the integration was only needed at a higher level, when monitoring data and results from modeling were finally integrated and a risk assessment was possible. These aspects are already treated in other paper cited in the manuscript (Versace et al., 2012; Costanzo et al., 2015; 2016). In this paper, we only deal with the interaction between GB-InSAR and the DCPC. In fact, what happen next (e.g. data integration between GB-InSAR and other instruments) falls out of our interest and detailed knowledge.**

Following, some specific comments and minor points to improve the text:

- Page 2, row 56: please replace "aerial" with "spatial"

**Done.**

- Page 2, rows 56-57: please replace "The installation was in an area where the only internet connection available 57 was 3G" with "the monitoring area was covered by a 3G mobile telecommunication networks"

**This sentence has been changed as suggested and it is now at the beginning of section 5, following the suggestion of referee #1.**

- Page 2, rows 61-83: these lines are very specific and of little interest to readers. Please consider deleting these lines or inserting them in Section 3.3.

**They have been deleted and only in small part moved.**

- Page 2, row 76: please change "where" with "were"

**This part has been removed from the paper.**

- Page 3, row 86: please change "ground-based interferometric synthetic aperture radar" with "Ground-Based Interferometric Synthetic Aperture Radar (GB-InSAR)"

**We have now changed this.**

- Section 3.1: this section appears too long. Please consider reducing sentences and adding a table with the technical specifications of equipment used (eg. Frequency, Bandwidth, Range and Cross-range resolution, etc.).

**Thank you for this comment. In fact this part was mostly a repetition and has been largely reduced. We also remove the part explaining the technical specifications of the equipment, since this information can be found in the literature cited in the paper and is not fundamental for our purpose.**

- Section 4: please add more geological information (eg. materials involved) to better frame the area under study.

**Now we have explained that "The lithologies outcropping in this area are Pliocene-Quaternary clay, clayey marlstones, and more recent (Holocene) terraced alluvial sediments (from clay to gravel). The landslides shown in Figure 2 are all located in clay or clayey marlstones".**

- Page 8, row 250: please add space before "These"

**Done.**

- Fig. 1: please improve the quality of figure

**Now the text is bolder and the boxes are no longer filled with a gradient but with a solid color.**

- Fig. 3: please increase the font size

**The font size has been increased.**

- Fig. 8: please insert the location of GB-InSAR instrument installed.

You are right. The yellow asterisk in the left of the images represents the location of the GB-InSAR. This is now specified in the caption of the image.